# Boosting Audio Visual Question Answering via Key Semantic-Aware Cues

## ABSTRACT

The Audio Visual Question Answering (AVQA) task aims to answer questions related to various visual objects, sounds, and their interactions in videos. Such naturally multimodal videos contain rich and complex dynamic audio-visual components, with only a portion of them closely related to the given questions. Hence, effectively perceiving audio-visual cues relevant to the given questions is crucial for correctly answering them. In this paper, we propose a **T**emporal-**S**patial **P**erception **M**odel (**TSPM**), which aims to empower the model to perceive key visual and auditory cues related to the questions. Specifically, considering the challenge of aligning non-declarative questions and visual representations into the same semantic space using visual-language pretrained models, we construct declarative sentence prompts derived from the question template, to assist the *temporal perception module* in better identifying critical segments relevant to the questions. Subsequently, a *spatial perception module* is designed to merge visual tokens from selected segments to highlight key latent targets, followed by cross-modal interaction with audio to perceive potential sound-aware areas. Finally, the significant temporal-spatial cues from these modules are integrated for answering the question. Extensive experiments on multiple AVQA benchmarks demonstrate that our framework excels not only in understanding audio-visual scenes but also answering complex questions effectively.

## CCS CONCEPTS

• **Computing methodologies** → **Scene understanding**.

## KEYWORDS

Multi-modal sense understanding, Audio visual question answering

## 1 INTRODUCTION

Audio and visual cues abundantly contribute to conveying information in our daily lives, and both modalities jointly improve our ability in scene perception and understanding [34]. For instance, imagining that we are driving along a winding mountain road, honking the horn ahead of time is often safer than relying solely on observing the road ahead with our eyes. In recent years, we have seen significant progress in sound source localization [12, 31]

**Figure 1: Identifying key temporal segments and spatial sound-aware areas is critical for fine-gained audio-visual scene understanding through human-like cognitive processes. For instance, in a scenario of violin and flute ensemble, regarding a given complex question: a) directly utilizing the question makes it difficult to effectively select key temporal segments; b) the lack of spatial supervision signals leads to challenges in capturing audio-visual association; c) our method, employing constructed declarative prompts, can accurately locate critical temporal segments and spatial sound-aware areas.**

and separation [9, 39, 40], event localization [3, 33], video parsing [26, 28, 32], segmentation [20, 24, 41], dialog [1, 30], question answering [6, 19, 35, 37], *etc.*, towards audio-visual scene understanding. Particularly, the Audio-Visual Question Answering (AVQA) task, involving the fine-gained spatio-temporal perception and reasoning of complex audio-visual scenes, has emerged as valuable and challenging focus of research interest.

To solve above AVQA task, Li *et al.* [19] build a large-scale MUSIC-AVQA dataset as a strong benchmark and propose a spatio-temporal grounding model to achieve scene understanding and reasoning over audio and visual modalities. Yun *et al.* [37] and Yang *et al.* [35] also introduce the Pano-AVQA and AVQA dataset to explore panoramic and real-life scene, respectively. Recently, LAVISH [23] introduced a novel parameter-efficient framework for encoding audio-visual scenes using off-the-shelf pretrained vision transformers, achieving notable progress. Moreover, researchers [5, 18] have

considered the significance of the given question, attempting to achieve precise perception of relevant temporal segments and spatial sound sources using the question as a guiding factor, leading to promising results. Clearly, these endeavors have markedly propelled the progress of research in audio-visual question answering.

Despite the significant progress made in AVQA, there are still several challenges that need to be addressed. **For temporal perception**, this task involves understanding long audio-visual videos, which suffer from heavy information redundancy. Existing works [19, 23, 27] typically employ a uniform sampling strategy to reduce redundancy and computational costs, but may lead to the loss of crucial information. Others [5, 15, 18] attempt to use the CLIP pre-trained model as a feature extractor by measuring the similarity between given questions and video frames to select temporally relevant segments. However, the given question's expression format is not declarative, inconsistent with the textual format used in the CLIP model, making it difficult to effectively align with the semantic content of video frames, and thus challenging the search for temporally relevant segments related to the input question. **Regarding spatial perception**, the lack of supervised information for spatial visual objects and sound makes it challenging for models to associate visual targets with sounds in the video, thereby making it difficult to identify potential sound-aware areas. While existing pre-trained object detection models used in visual question answering task can excel in key object localization, the absence of certain specific categories (*eg*., suona, guzheng, *etc*.) in the AVQA-related datasets, adding difficulty to locate relevant areas. Some explorations [18] have employed ViT to convert video frames into token sequences, utilizing semantic similarity calculations between questions and tokens to perceive the most relevant tokens as key targets. Nevertheless, the absence of objects' semantics in visual tokens makes establishing an effective correlation with questions challenging, thus rendering accurate localization difficult. This imprecise spatial perception makes it difficult to establish effective associations with sounds in the video, thereby complicating the localization of potential sound-aware areas. Hence, as shown in Fig. 1, it is crucial to enable machines to perceive complex audio-visual scenes in a manner akin to human cognition and accurately infer answers to questions.

To address these challenges, we propose an effective Temporal-Spatial Perception Model (TSPM) to perceiving crucial visual and auditory cues related to the questions in complex audio-visual scenes. **Firstly**, the content related to the question is usually scattered in partial segments of the video instead of the whole sequence. Hence, we design a Text Prompt Constructor (TPC), which construct a declarative sentence text prompt derived from the question template, effectively aligning it with the semantic content of the visual frames. Following this, we introduce a Temporal Perception Module (TPM) that utilizes cross-modal attention mechanisms to identify the key temporal segments relevant to the given question. **Secondly**, identifying key visual areas and their corresponding sound source positions within critical segments, which can help to learn audio-visual associations in complex scenarios. To achieve this, the Spatial Perception Module (SPM) is designed to merge visual tokens on selected temporal segments to multiple joint tokens, thus preserving the semantic information of potential targets. Then, these joint tokens interact cross-modal interaction with audio to

perceive potential sound-aware areas. **Finally**, the above critical temporal segments and sound-aware regions' features are fused to obtain a joint representation for question answering. Extensive experiments on multiple benchmark demonstrate that our proposed approach achieves precise temporal-spatial perception, highlighting its immense potential in tackling audiovisual question-answering task. Our contributions can be summarized as follows:

- The temporal perception module designed in TSPM transforms questions into declarative prompts using a constructed declarative sentence generator, facilitating better alignment with the semantic of visual frames and effectively identifying key temporal segments relevant to the given question.
- The spatial perception module introduced in TSPM merges visual tokens on selected temporal segments to preserve key potential targets, then engages in cross-modal interaction with audio, effectively perceiving potential sound-aware areas.
- Extensive experiments on multiple benchmarks demonstrate that the proposed TSPM achieves precise spatio-temporal perception, showcasing its significant potential in addressing audio-visual question answering task.

## 2 RELATED WORKS

### 2.1 Audio Visual Scene Understanding

Inspired by the multisensory perception of humans, the community has paid more and more attention to audio-visual scene understanding in recent years. [34]. Compared to other modalities, visual and auditory modalities possess unique characteristics such as cognitive foundation, semantic consistency, spatial consistency, temporal consistency, and rich support from real-world data. It includes various interesting tasks such as sound source localization [12, 13, 31], action recognition [10], event localization [3, 33], video parsing [26, 28, 32], segmentation [20, 24, 41], dialog [1, 30], *etc*. These studies integrate rich audiovisual information within multimodal scenes to overcome limitations in perception inherent to single modalities, thereby utilizing both auditory and visual modalities to explore finer-grained scene comprehension.

Apart from the above methods that facilitate scene understanding by excavating and analyzing different modalities, a unified multimodal model should also be able to reason their spatio-temporal correlation. Therefore, we focus on the audio-visual question answering task [6, 19, 35, 37] and explore spatiotemporal perception and reasoning in the audio-visual context.

### 2.2 Audio Visual Question Answering

Audio-visual question answering, which exploits the natural multimodal medium of video, is attracting increasing attention from researchers [4, 23]. It requires comprehensive understanding and integration of diverse modalities, leading to precise responses to distinct questions. To explore the above AVQA task, Yun *et al.* [37] proposed the Pano-AVQA, which includes 360-degree videos and their corresponding question-answer pairs, aimed at exploring understanding of panoramic scenes. Li *et al.* [19] presented that the MUSIC-AVQA has become a strong benchmark for promoting spatio-temporal reasoning research in dynamic and long-term audio-visual scenes. Considering that real-life scenarios contain

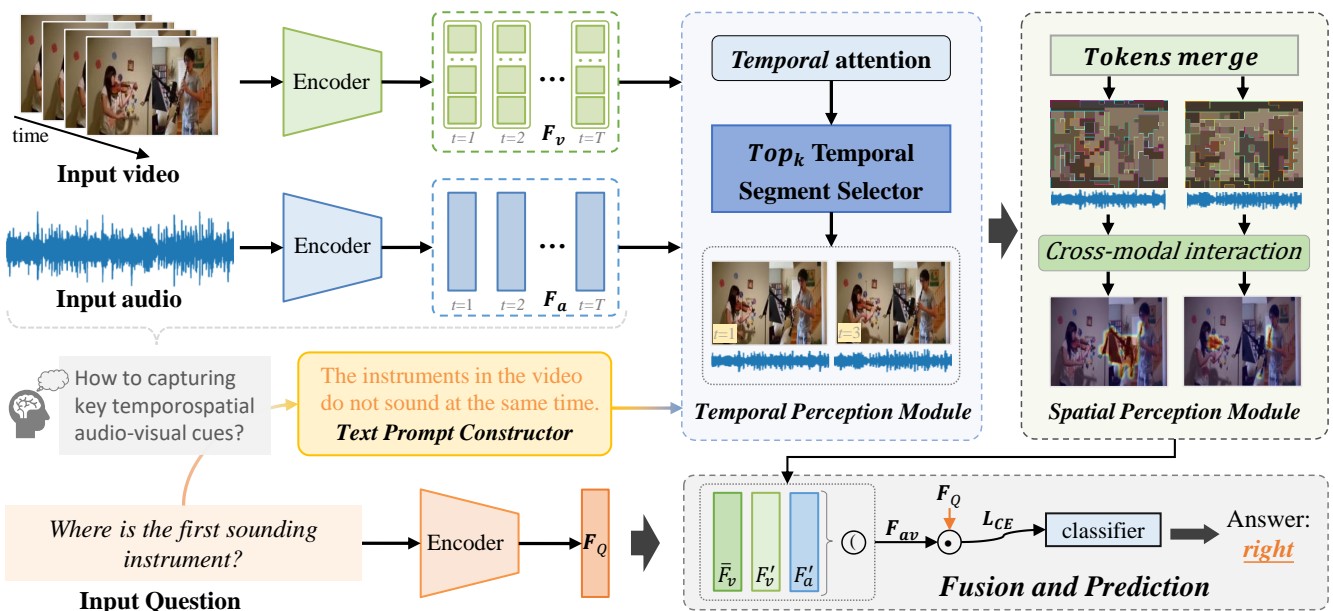

**Figure 2: Our proposed Temporal-Spatio Perception Model (TSPM) framework. Firstly, the video is divided into $T$ segments, and we use a pre-trained model to extract audio, visual, and question features. Then, a temporal perception module incorporating constructed prompt aiming to effectively capturing $Top_k$ key relevant temporal segments. Subsequently, the spatial perception module is designed to enhance spatial awareness through the interaction of audio-visual tokens.**

greater variety of audio-visual daily activities, AVQA benchmark is proposed in [35], which further expands the audio visual scene coverage of AVQA task. Recently, LAVISH [23] have dedicated to exploring improvements in audio-visual association and enhancing training efficiency, resulting in satisfactory outcomes.

Above research extract audio and visual features globally, without considering the importance of local feature representation. Chen *et al.* [5] consider the importance of the given question, which guides the feature extraction of both audio and visual signals. And then the PSTP-Net [18] is proposed explore critical temporal segments and sound-aware regions among the complex audiovisual scenarios progressively. However, aligning questions with video semantics is challenging due to the non-declarative nature, making it hard to identify key relevant segments. Our work focuses on empowering the model to gradually perceive essential visual and auditory cues for audio-visual scene understanding.

## 3 METHOD

To solve the AVQA challenges, we propose an effective Temporal-Spatial Perception Model (TSPM) to achieve fine-gained audio-visual scene understanding, thus answering questions accurately. An overview of the proposed framework is illustrated in Fig. 2.

### 3.1 Input Representation

Given an input audio-visaul video sequence, we first divide it into $T$ non-overlapping audio and visual segment pairs $\{a_t, v_t\}_{t=1}^{T}$, where each segment is 1s long. Subsequently, we partition each visual frame into $M$ patches and append a special [CLS] token to the beginning of the first patch. The question sentence $Q$ is tokenized into $N$ individual words $\{q_n\}_{n=1}^{N}$.

**Audio Representation**. For each audio segment $a_t$, we use the pre-trained VGGish [11] model to extract the audio feature as $f_a^t \in \mathbb{R}^D$, where $D$ is the feature dimension. The pretrained VGGish model is a VGG-like 2-D CNN network that trained on the large-scale AudioSet [11] dataset, employing over transformed audio spectorgrams. Then the features at the audio spectrgram second-level can be interpreted as $F_a = \{f_a^1, f_a^2, ..., f_a^T\}$.

**Visual Representation**. A fixed number of frames are sampled from each visual segment $v_t$. Then we apply pre-trained CLIP [29], with frozen parameters, extract both frame-level and token-level features as $f_v^t$ and $f_p^t$ on video frames, respectively, where $f_v^t \in \mathbb{R}^D$, $f_p^t \in \mathbb{R}^{M \times D}$ and $M$ are token numbers of one frame. Finally, the visual frame-level and token-level features can be denoted as $F_v = \{f_v^1, f_v^2, ..., f_v^T\}$, $F_p = \{f_p^1, f_p^2, ..., f_p^T\}$, respectively.

**Text Representation**. Given an asked question $Q$, we represent each word $q_n$ in a fixed length vector with word embeddings, and then feed it into the pre-trained CLIP[29] model to get the question feature $F_Q$, where $F_Q \in \mathbb{R}^D$. Note that the first token pooling is used for extracting question features.

### 3.2 Temporal Perception Module

To hightlight the $Top_k$ crucial temporal segments that relevant to the question, we propose a **T**emporal **P**erception **M**odule (**TPM**) with a carefully designed text prompt. While previous works [5, 18] have considered identifying key segments through the semantic similarity between question and temporal visual segments, aligning questions with visual frame semantics poses a significant challenge due to the non-declarative sentence of the questions. Therefore, the key of TPM lies in constructing a declarative sentence, aligning it

**Algorithm 1:** $Top_k$ Segments Selection Operation.

**Input:** Visual feature $F_v$ and its temporal attention weights: $W$; audio feature $F_a$, $Top_k$ numbers;
**Output:** Selected $Top_k$ segment features $F_a'$, $F_v'$ and its index $\Omega_{TPM}$;

Init list $\Omega_{TPM}$, $F_v'$ for selected index and feature;
**if** $cnt < Top_k$ **then**
  $\quad \Omega_{TPM} \leftarrow \text{Index}[\arg \max W]$
  $\quad W \leftarrow W - \arg \max W$;
  $\quad cnt \leftarrow cnt + 1$;
  $\quad$ Update $\Omega_{TPM}$, $W$;
**end**
**for** $idx$ $in$ $\Omega_{TPM}$ **do**
  $\quad F_v' \leftarrow idx\text{-th } F_v$; $F_a' \leftarrow idx\text{-th } F_v$;
  $\quad$ Update $F_a'$, $F_v'$, $\Omega_{TPM}$; Minus the current idx
**end**

effectively with the semantic content of the video, and facilitating the identification of critical segments.

To achieve this, we devised a **T**ext **P**rompt **C**onstructor (**TPC**) with the goal of generating declarative statements based on input questions. This help semantic alignment between the generated statements and the visual frame, facilitating to identify key temporal segments to enhance the model's temporal perception ability. Since the input question do not contain answers, directly transforming them into declarative statements pose difficulties. Hence, we approached the problem from a new perspective, considering the design of statements that exclude irrelevant segments, guiding the model's attention toward temporal content relevant to the questions. Illustrated by the example in Fig. 2, for input question "*Where is the first sounding instrument?*", the objective is to identify the moment when the *first* instrument starts playing. Considering that instruments in the video do not play simultaneously but follow a sequential order, we direct the model's attention to segment in the video where instruments do not play simultaneously. This way directs the model focus on segments where there are changes in the order of instrument sounds, identifying crucial segments. Leveraging the **TPC**, we manually transform the question into a declarative sentence "*The instruments in the video do not sound at the same time.*", denote as TPrompt, aligning its feature representation well with the semantic content of the video. This allows us to locate segments related to TPrompt and subsequently locate temporal segments relevant to the question.

For a given declarative sentence TPrompt, its feature embedding $F_{TPrompt}$ using the same encoder as the given question. Concretely, we firstly use one linear projection layer $\text{Key}(\cdot)$ to transform indexing visual features $F_v$ to indexing keys $\mathbf{k}$. Then we get an attention score for each indexing key in the video temporal sequence. A Softmax layer normalizes the attention scores and generates an attention weight vector $W$ by:

$$W = \text{Softmax}(\frac{F_{TPrompt} \cdot [\mathbf{k}_1, \mathbf{k}_2, ..., \mathbf{k}_T]^\top}{\sqrt{d}}), \qquad (1)$$

where $\mathbf{k}_j = \text{Key}(F_v^j)$, $j \in \{1, 2, ..., T\}$, and $d$ is the dimensionality of the key vector. Considering that the higher weight indicates a

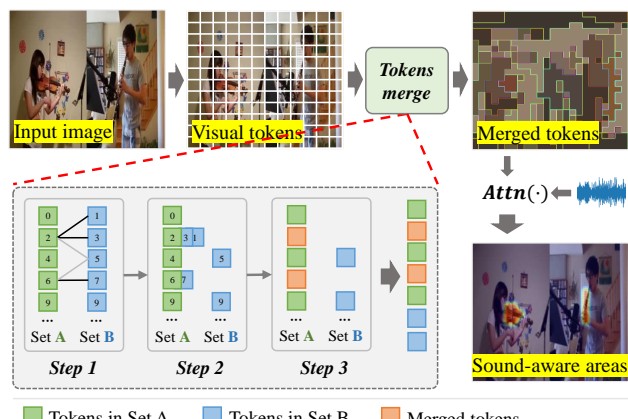

Figure 3: Spatial Perception Module. Similar tokens are merged. For example, for a given complex sense, the man is playing flute if merged into a single token, and the woman is playing violin is merged into a single token. Following this, the proposed model identify the *sounding* instrument, thus inferring the correct answer to the input question.

stronger correlation between the video content and TPrompt, we conduct $Top_k$ feature selection over $T$ segments. To be specific, we employ a temporal selection operation algorithm, denoted as $\Psi$, which is implemented by the sorted algorithm for ranking and sorting to pick out the crucial relevant segments with the highest attention weights and their corresponding indices:

$$F_a', F_v', \Omega_{TPM} = \Psi(F_a, F_v, W, Top_k), \qquad (2)$$

where $\Psi$ is shown in *Algorithm 1*, $\Omega_{TPM}$ is the index position corresponding to the $Top_k$ highest weights, $\Omega_{TPM} \in \{0, 1, ..., k-1\}^{Top_k}$, $F_a'$, $F_v'$ is selected temporal feature, $F_a' \in \mathbb{R}^{Top_k \times D}$, $F_v' \in \mathbb{R}^{Top_k \times D}$. Note that the $Top_k$ temporal audio segments are corresponding to positions on the $Top_k$ visual segments relevant to the question.

## 3.3 Spatial Perception Module

To identify visual regions that are pertinent to the key instrument, the **S**patial **P**erception **M**odule (**SPM**) is designed to merge visual tokens in selected temporal segments based on similarity, preserving their semantics, and subsequently engages in cross-modal interaction with audio to enhance audio-visual association. Given previous works [18] attempt to identify crucial regions by leveraging the semantic similarity between questions and visual tokens, the lack of semantic about objects within these tokens presents a challenge in establishing effective correlations with sound.

To address this, we enhanced the preservation of semantic information in visual tokens along selected key temporal sequences. We achieve this by merging similar tokens within each visual frame, resulting in merged tokens that carry richer semantic information about objects. Specially, given the visual token-level embedding $F_p$ and $Top_k$ curious temporal segment index, we obtain the temporal visual token-level features as follow:

$$F_p' = \Phi(F_p, \Omega_{TPM}), \qquad (3)$$

where $F_p' \in \mathbb{R}^{Top_k \times M \times D}$. and $\Phi$ the algorithm that determines the temporal position in $F_p$ based on the index in $\Omega_{TPM}$. For the given

selected visual token-level feature $F'_p$, we employ a token-merging strategy to enhance the semantic features between the attention and MLP branches of each transformer block. Then, similar tokens are merged in each transformer block with per layer, and the merged visual token-level feature $\hat{F}_p$ as:

$$\hat{F}_p = \mathbf{Merge}(F'_p), \tag{4}$$

where $\hat{F}_p = \{\hat{f}_p^1, \hat{f}_p^2, ..., \hat{f}_p^\lambda\}$, $\hat{F}_p \in \mathbb{R}^{\lambda \times S \times D}$ and $\lambda$ is selected temporal segments' moment, $S$ is merged tokens number. Specifically, as shown in Fig. 3, evenly divide the $M$ tokens in $F'_p$ into two subsets $A$ and $B$ of roughly equal size in *Step 1*. Then, for one subset $A$, calculating the similarity between each token and every token in the other subset $B$, drawing an edge for each calculated similarity. Subsequently, apply mean fusion to the tokens connected by the similar edges. And in *Step 2*, concatenating the two subsets to generate a merged visual token-level feature $\hat{F}_p$. It's worth noting that between each transformer block's attention branch and MLP branch, *Step 1* through *Step 3* of the visual token merging process are executed, resulting in the creation of multiple merged tokens with semantic representations.

Then, considering that the sound and the location of its visual source usually reflects the patial association between audio and visual modality, we leverage the powerful cross-modal perception ability to interact between selected visual merged token-level features and audio embeddings $\hat{F}_p, F'_a$. This enables concrete audio-visual correlation which performs attention-based patch-level merged tokens sound source perception. Denote $\mathtt{Attn}(\cdot)$ to be the scaled dot-product conducted on the query, keys, and values, the aggregated feature can be obtained by:

$$\overline{F}_v = \hat{f}_p^\lambda + \mathtt{Attn}(\hat{f}_p^\lambda, \hat{F}_p, \hat{F}_p) + \mathtt{Attn}(f_a^\lambda, \hat{F}_p, \hat{F}_p), \tag{5}$$

where $\overline{F}_p \in \mathbb{R}^{Top_k \times S \times D}$. Thus far, we have progressively identified the key temporal segments that are most relevant to the input question, and its potential sound-aware areas.

## 3.4 Multimodal Fusion and Answer Prediction

To achieve AVQA task, we concatenate the updated visual features $F'_v, \overline{F}_v$, and the audio features $F'_a$ obtained from TPM and SPM, respectively. Then the visual fusion feature $F_{av}$ obtained by a linear layer. To verify the audio-visual fusion of our proposed effective Temporal-Spatial Perception Model, we employ a simple element-wise multiplication operation to integrate the question feature $F_q$ and the previously obtained audio-visual fusion embedding $F_{av}$. And it can be formulated as:

$$F_{av} = FC(Concat[F'_a, F'_v, \overline{F}_v]), \tag{6}$$

$$e = F_q \odot F_{av}, \tag{7}$$

where $\odot$ is element-wise multiplication operation, $\delta$ and $FC$ represent $Tanh$ activation function and linear layer, respectively. Then a $\mathtt{Softmax}$ function is used to output probabilities $p \in \mathbb{R}^C$ for candidate answers, where $C$ is the size of the pre-defined candidate answer vocabulary pool. With the predicted probability vector and the corresponding groundtruth label y, we optimize it using a cross-entropy loss: $\mathcal{L}_{qa} = -\sum_{c=1}^C y_c log(p_c)$. During testing, we can select the predicted answer by $\hat{c} = \arg\max_c(p)$.

## 4 EXPERIMENTS

### 4.1 Datasets

**MUSIC-AVQA** [19], it contains 9,288 videos covering 22 different musical instruments, with a total duration of over 150 hours and 45,867 question-answering pairs. The questions are designed under multi-modal scenes containing 33 question templates covering nine types, i.e., the audio-visual, separate visual, and separate audio, depending on which modalities are used to discover question-related clues for answer prediction. The diversity question-answering pairs which occupy a large portion of the entire dataset, and there are five audio-visual question types referring to *existential*, *counting*, *location*, *comparative*, and *temporal*. The large-scale MUSIC-AVQA dataset is well suited for studying temporal-spatial perception for dynamic and long-term audio-visual scenes.

**AVQA** [35], is designed for audio-visual question answering in general real-life scenario videos. It contains 57,015 videos from daily audio-visual activities, along with 57,335 question answering pairs designed relying on clues from both modalities, where information from a single modality is insufficient or ambiguous.

For both datasets, we adopt the official split of the two benchmarks into training, evaluation, and test sets.

### 4.2 Implementation Details

For the visual stream, we divide the video into 1-second segments and frames sample rate at $1fps$. We utilize the CLIP-ViT-L/14 [29] model pre-trained on ImageNet to extract 512-$D$ feature representations for each visual segment, where $[CLS]$ token denote visual frame-level features. For the audio signal, it is sampled at 16kHz, which is a standard sampling rate for audio. we use the VGGish network pre-trained on AudioSet to extract 128-$D$ features. For each input question sentence, we extract its feature same as visual frame-level encoder to obtain 512-$D$ feature vector. In all experiments, we use Adam optimizer with an initial learning rate of 1$e$-4 and will drop by multiplying 0.1 every 10 epochs. The batch size and number of epochs are set to 64 and 30, respectively. The constructed statement templates are provided in the *supplementary code* files. We use the *thop* library in PyTorch to calculate the model's parameters and FLOPs. Our proposed model is trained on NVIDIA GeForce RTX 3090 and implemented in PyTorch.

### 4.3 Quantitative Results and Analysis

To verify the effectiveness of the proposed TSPM, we compare it with multiple existing mehtods: AVSD [30], Pano-AVQA [37], AVST [19], LAVISH [23], COCA [16], PSTP-Net [18], *etc.* Tab. 1 indicate that the TSPM outperforms all comparison methods. Specially, the TSPM method shows significant improvements in the subtask types of *Audio-visual*, including *Localization*, and *Temporal*. Specifically, compared to the recent PSTP-Net [18], the model achieves remarkable improvements of 0.92% (73.51% and 71.26%) in the above-mentioned complex audio-visual question types. It is worth noting that the model shows a performance boost of 5.14% (82.29% and 77.15%) and 7.54% (84.90% and 77.36%) in the *Counting* and *Localization* subtasks of the visual modality, respectively, when compared to PSTP-Net [18]. The significant performance improvements indicate that our TSPM is effectively identifies crucial

| Method | Audio | | | Visual | | | Audio-Visual | | | | | | Avg |
|---|---|---|---|---|---|---|---|---|---|---|---|---|---|
| | Count | Comp | Avg | Count | Local | Avg | Exist | Count | Local | Comp | Temp | Avg | |
| FCNLSTM [8] | 70.80 | 65.66 | 68.90 | 64.58 | 48.08 | 56.23 | 82.29 | 59.92 | 46.20 | 62.94 | 47.45 | 60.42 | 60.81 |
| GRU [2] | 71.29 | 63.13 | 68.28 | 66.08 | 68.08 | 67.09 | 80.67 | 61.03 | 51.74 | 62.85 | 57.79 | 63.03 | 65.03 |
| BiLSTM [42] | 67.75 | 63.64 | 66.23 | 59.65 | 29.80 | 44.55 | 80.97 | 56.05 | 36.09 | 62.67 | 32.60 | 54.93 | 54.17 |
| Hco_Att [25] | 70.80 | 54.71 | 64.87 | 63.49 | 67.10 | 65.32 | 79.48 | 59.84 | 48.80 | 56.31 | 56.33 | 60.32 | 62.45 |
| MCAN [36] | 78.07 | 57.74 | 70.58 | 71.76 | 71.76 | 71.76 | 80.77 | 65.22 | 54.57 | 56.77 | 46.84 | 61.52 | 65.83 |
| PSAC [22] | 75.02 | 66.84 | 72.00 | 68.00 | 70.78 | 69.41 | 79.76 | 61.66 | 55.22 | 61.13 | 59.85 | 63.60 | 66.62 |
| HME [7] | 73.65 | 63.74 | 69.89 | 67.42 | 70.20 | 68.83 | 80.87 | 63.64 | 54.89 | 63.03 | 60.58 | 64.78 | 66.75 |
| HCRN [17] | 71.29 | 50.67 | 63.69 | 65.33 | 64.98 | 65.15 | 54.15 | 53.28 | 41.74 | 51.04 | 46.72 | 49.82 | 56.34 |
| AVSD [30] | 72.47 | 62.46 | 68.78 | 66.00 | 74.53 | 70.31 | 80.77 | 64.03 | 57.93 | 62.85 | 61.07 | 65.44 | 67.32 |
| PanoAVQA [37] | 75.71 | 65.99 | 72.13 | 70.51 | 75.76 | 73.16 | 82.09 | 65.38 | 61.30 | 63.67 | 62.04 | 66.97 | 69.53 |
| ST-AVQA [19] | 77.78 | _67.17_ | 73.87 | 73.52 | 75.27 | 74.40 | _82.49_ | 69.88 | 64.24 | 64.67 | 65.82 | 69.53 | 71.59 |
| COCA [16] | 79.35 | **67.68** | 75.42 | 75.10 | 75.43 | 75.23 | **83.50** | 66.63 | 69.72 | 64.12 | 65.57 | 69.96 | 72.33 |
| PSTP-Net [18] | 73.97 | 65.59 | 70.91 | 77.15 | 77.36 | 77.26 | 76.18 | 72.23 | _71.80_ | **71.79** | _69.00_ | _72.57_ | 73.52 |
| LAVISH [23] | _82.09_ | 65.56 | _75.97_ | _78.98_ | _81.43_ | _80.22_ | 81.71 | _75.51_ | 66.13 | 63.77 | 67.96 | 71.26 | _74.46_ |
| **TSPM** (Ours) | **84.07** | 64.65 | **76.91** | **82.29** | **84.90** | **83.61** | 82.19 | **76.21** | **71.85** | _65.76_ | **71.17** | **73.51** | **76.79** |

Table 1: Temporal-Spatial Perception Model results on the test set of MUSIC-AVQA. The top-2 results are highlighted.

| Method | Audio | Visual | Audio-Visual | Avg |
|---|---|---|---|---|
| AVSD [30] | 71.74 | 69.21 | 65.41 | 67.53 |
| ST-AVQA [19] | 71.74 | 70.71 | 66.53 | 68.56 |
| LAVISH [23] | 73.00 | _77.62_ | _70.09_ | _72.60_ |
| PSTP-Net [18] | _73.58_ | 76.44 | 67.66 | 71.06 |
| **TSPM** (Ours) | **76.93** | **81.07** | **71.93** | **75.27** |

Table 2: TSPM results on the test set of MUSIC-AVQA (split by video id). The top-2 results are highlighted.

| Method | Audio | Visual | Audio-Visual | Avg |
|---|---|---|---|---|
| _w/o._ all | 73.93 | 79.23 | 70.37 | 73.35 |
| _w/o._ TPM | 75.85 | 82.74 | 72.53 | 75.82 |
| _w/o._ SPM | 77.16 | 81.92 | 72.25 | 75.68 |
| _w/o._ TPC | 75.54 | 82.20 | 72.96 | 75.87 |
| _w/._ QPrompt | 76.47 | 81.30 | 71.82 | 75.16 |
| _w/o._ Merge | 75.79 | 82.91 | 72.84 | 76.03 |
| **TSPM** (Ours) | 76.91 | 83.61 | 73.51 | 76.79 |

Table 3: TSPM's module configuration results.

temporal segments and spatio tokens in videos. Moreover, in comparison to LAVISH [23], which fine-tunes large pretrained models, our model demonstrates superior efficiency without the need for fine-tuning. We conducted tests with an equal number of epochs under the same hardware configuration, and it was observed that LAVISH incurred a cost 14× higher than our model. Additionally, we observed limitations in the performance of *Comparative* type questions, and we consider this may be attributed to the challenges of separating multiple sound in complex audio-visual scenes. This motivates us to explore strategies (such as dynamic fusion) in future work that can achieve better performance on both single-modality and multi-modality aspects.

To further validate the capabilities of the proposed TSPM, in contrast to splitting by *Question ID*, we also partitioned the MUSIC-AVQA dataset based on *Video ID*, apportioning it into training, validation, and test sets in a ratio of 7 : 1 : 2. As shown in Tab. 2, it can be seen that our proposed TSPM achieves the best overall performance compared to the latest AVQA methods. Particularly, in the three subtask types of audio, visual, and audio-visual, our TSPM outperforms others significantly, showcasing the excellent generalization capability and performance of the proposed TSPM.

In summary, the TSPM offers significant improvements over existing approaches and provides a novel insight into question-oriented audio-visual scene understanding.

## 4.4 Ablation Studies

In this subsection, we delve into examining the impact of various modules within the TSPM on the performance of the MUSIC-AVQA.

To verify the effectiveness of the proposed components, *i.e.*, TPM, SPM, TPC, Tokens merge, *etc.*, we remove them from the primary model and re-evaluate the new model's performance. Tab. 3 shows that after removing a single component, the overall model's performance decreases, and different modules have different performance effects. The specific analysis is as follows:

- **TSPM _w/o._ all**. When we remove all designed modules or components within the framework, retaining only the simple fusion operation of input audio, video, and question features, a significant decline (76.79% and 73.35%) in model performance can be clearly observed from Tab. 3. This pronounced deterioration serves as compelling evidence that the multiple components intricately designed within the proposed TSPM play a pivotal role in bolstering the model's overall effectiveness.

- **TSPM _w/o._ TPM**. The motivation behind designing the TPM is to enable the model to select temporal segments most relevant to the given question. To validate the necessity of the TPM, we removed the TPM from the TSPM and assessed the performance of the new model. As shown in Tab. 3, when the TPM was removed, the new model's performance decreased to 75.82%, representing a 0.97% decrease compared to when TPM was utilized. Furthermore, noticeable performance declines were observed across the audio, visual, and audio-visual subtask types. These experimental results underscore the importance of TPM, which effectively enables the model to perceive crucial temporal segments, thereby enhancing temporal perception performance.

| Method | Audio | Visual | Audio-visual | Avg |
|---|---|---|---|---|
| Top-$k$=10 | 76.91 | 83.61 | 73.51 | 76.79 |
| Top-$k$=20 | 76.60 | 82.58 | 73.41 | 76.40 |
| Top-$k$=30 | 77.41 | 82.70 | 73.55 | 76.66 |
| Top-$k$=40 | 76.41 | 82.78 | 72.94 | 76.16 |
| tokens=8 | 75.54 | 82.00 | 73.08 | 75.88 |
| tokens=14 | 76.91 | 83.61 | 73.51 | 76.79 |
| tokens=27 | 76.16 | 82.20 | 73.00 | 76.00 |
| **TSPM** (Ours) | 76.91 | 83.61 | 73.51 | 76.79 |

Table 4: Effects of TSPM's parameter configuration.

| Method | Training Param (M) | FLOPs (G) | Acc (%) |
|---|---|---|---|
| ST-AVQA [19] | 18.48 | 3.19 | 71.59 |
| PSTP-Net [18] | 4.30 | 1.22 | 73.52 |
| LAVISH [23] | 21.09 | – | 74.46 |
| **TSPM** (Ours) | 6.22 | 1.42 | 76.79 |

Table 5: Parameter and FLOPs.

- **TSPM *w/o.* SPM**. The purpose of the SPM is to identify key objects and potential sound-aware areas within the visual frame. To demonstrate the significance of the SPM, we conducted an experiment where it is removed. As shown in Tab. 3, compared with TSPM, the result decreased to by 1.11% (from 76.79% to 75.68%), indicating the importance of the spatial perception in improving performance.

- **TSPM *w/o.* TPC**. The designed TPC primarily generates declarative sentence text based on the given question, aligning it with the semantic of video frames to better identify temporal segments relevant to the question. Removing this module implies that all video frames (T=60) will be selected, potentially leading to temporal redundancy. As observed in Tab. 3, the utilization of TPC effectively enhances model performance, resulting in a 1.12% improvement (from 75.87% to 76.79%), thereby strengthening temporal perception capability.

- **TSPM *w/o.* QPrompt**. To validate whether transforming the given question into declarative statement indeed leads to better selection of key temporal segments, thereby effectively improving model performance, we replaced the constructed statements with input questions. As shown in Tab. 3, in this scenario, the model's performance is significantly lower compared to when using declarative statements (76.79% and 75.16%). The experimental results demonstrate the necessity of using declarative statements and indirectly highlight the importance of TPC.

- **TSPM *w/o.* Tokens merge**. Removing the Tokens merge operation from TSPM allows us to investigate whether it can preserve the semantic information of visual frame tokens. When this operation is removed, direct cross-modal interactions are conducted between all visual tokens and their corresponding temporal audio features. Tab. 3 shows that when Tokens merge is removed, there is a decrease in model performance, highlighting the importance of the Tokens merge strategy.

In general, each module contributes to the better performance. When all modules are present, the TSPM achieves best result on the MUSIC-AVQA dataset. Similarly, we explored the impact of key parameter configurations on model performance. As shown

| Method | Ensemble | Total Accuracy (%) |
|---|---|---|
| HME [7] | HAVF [35] | 85.0 |
| PSAC [22] | HAVF [35] | 87.4 |
| LADNet[21] | HAVF [35] | 84.1 |
| ACRTransformer [38] | HAVF [35] | 87.8 |
| HGA [14] | HAVF [35] | 87.7 |
| HCRN [17] | HAVF [35] | 89.0 |
| PSTP-Net [18] | – | 90.2 |
| TSPM *w/o.* all | – | 87.1 |
| TSPM *w/o.* TPM | – | 89.4 |
| TSPM *w/o.* SPM | – | 88.6 |
| **TSPM** (Ours) | – | **90.8** |

Table 6: TSPM results on the test of AVQA.

in Tab. 4, when the $top_k$ value is large, it may introduce temporal redundancy. When there are too many tokens, semantic merging on the token is not thorough enough; conversely, an excessive merging may result in semantic loss. The model achieves optimal performance when $top_k$ = 10 and $tokens$ = 14, respectively.

## 4.5 Computational costs

Tab. 5 illustrates the computational costs of TSPM compared with ST-AVQA [19], PSTP-Net [18] and LAVISH [23]. It can be observed that TSPM has fewer training parameters, lower FLOPs, and higher accuracy compared to ST-AVQA. Although PSTP-Net boasts lower computational costs, our TSPM achieves superior results at extremely low computational costs. LAVISH achieves a well accuracy, but its parameter is more than three times that TSPM. This is because LAVISH fine-tunes large pretrained models, whereas TSPM achieves comparable results without fine-tuning. In summary, our proposed TSPM achieves high performance at relatively low cost, fully demonstrating the effectiveness and efficiency of the model.

## 4.6 Experiments on AVQA dataset

To verify the generalization capability of the proposed TSPM, we compared it with multiple existing AVQA-based methods, including ACRTransformer [38], HCRN [17] , PSTP-Net [18], *etc.*, on the AVQA dataset. As illustrated in Tab. 6, the TSPM exhibits remarkable performance compared to recent methods. Specifically, our approach outperforms PSTP-Net [18] by 0.6% (90.8% and 90.6%), demonstrating notable superiority over earlier methods such as ACRTransformer [38]. Furthermore, while the performance improvement of TSPM on the AVQA dataset seems limited compared to its performance on the MUSIC-AVQA dataset, we attribute this primarily to the AVQA dataset's shorter duration ($10s$ . $60s$) and simpler audio-visual components.

Despite these differences, our TSPM maintains its effectiveness even in this scenario. Notably, in Tab. 6, the PSTP-Net [18] achieved a 1.2% improvement over the HCRN [17], while our TSPM exhibited a more substantial 1.8% enhancement, indicating the significant effectiveness of TSPM's performance boost. Additionally, ablation studies further confirm the effectiveness of both TPM and SPM components. Moreover, for the experimental settings on the AVQA dataset, we selected the Top-$k$ = 8 temporal segments relevant to the given question, with a merged token count of $tokens$ = 14. It's worth noting that HAVF [35] in Tab. 6, serving as the baseline

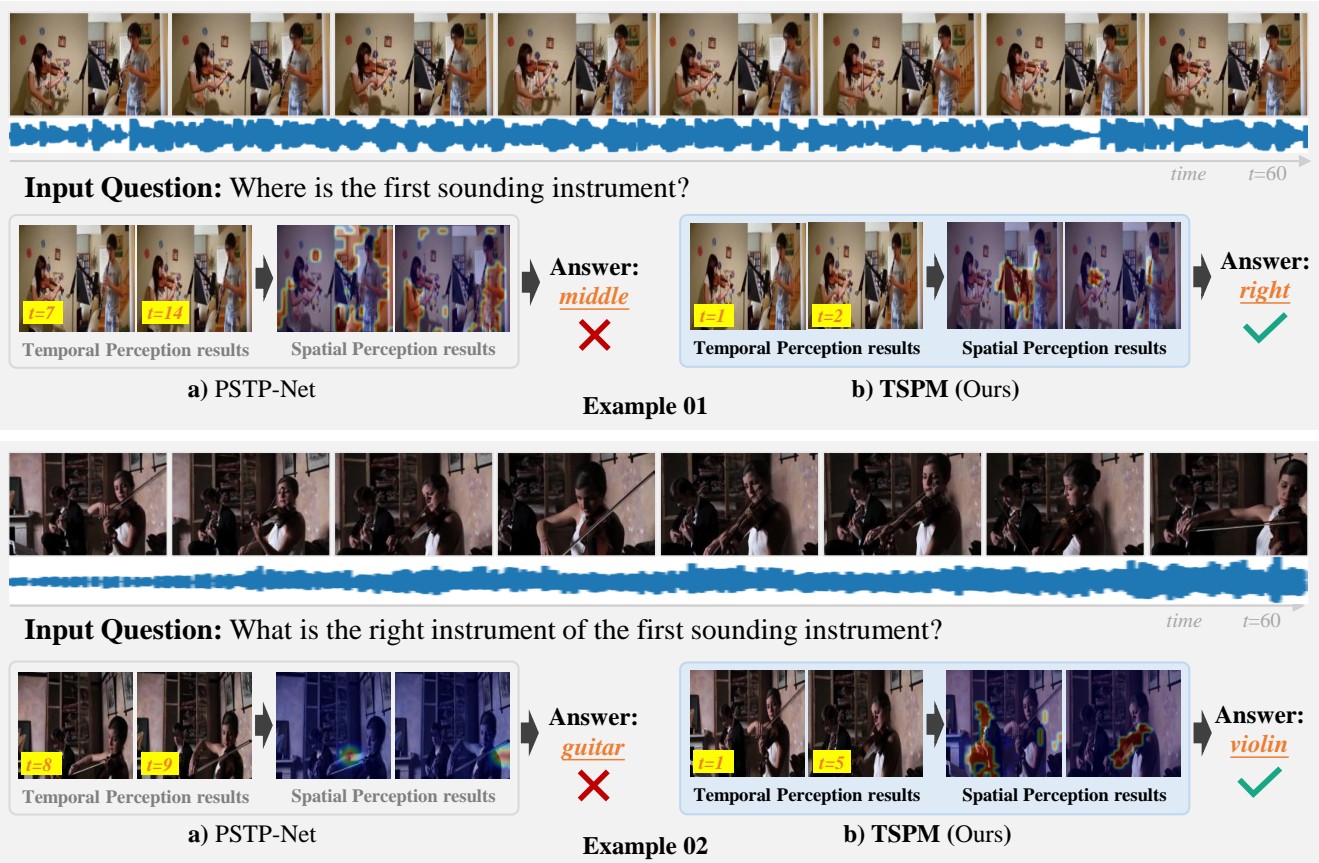

**Figure 4: Visualized TSPM results. In the showcased examples, we compared our proposed TSPM with the recent AVQA-related method PSTP-Net [18]. It can be observed that TSPM can progressively select relevant temporal segments and locate potential sound-aware areas, thus accurately providing correct answers to the given questions. This process vividly demonstrates TSPM's effective spatiotemporal perception capabilities in complex audiovisual scenarios.**

method for the AVQA dataset, includes three fusion modalities and integrates their outputs using an averaging strategy to generate answers. In summary, the proposed TSPM effectively demonstrates both its effectiveness and generalization.

### 4.7 Visualization Results

To showcase the temporal and spatial perception capabilities of the proposed TSPM, we provide two examples contrasting with the recent AVQA-related method PSTP-Net in Fig. 4. In *Example 01*, when presented with the question "*Where is the first sounding instrument?*", the TPM first identifies the temporal indices relevant to the question. Subsequently, the SPM sequentially locates potential sound-aware areas, with the heatmap indicating these regions. In this example, it becomes apparent that initially, only the "*flute*" on the *right* side is playing, but as time progresses, the violin on the left side also begins playing. The heatmap effectively illustrates the variation in multiple instruments playing within this dynamic and complex audio-visual scene. Consequently, it can be inferred that the correct answer to the question is the instrument on the "*right*" side. Similarly, in *Example 02*, the progressive temporal-spatial

perception process is aptly demonstrated, resulting in the correct answer. These visualizations indicate that the proposed TSPM can effectively perceive the temporal segments relevant to the question and the spatial areas associated with sound, showcasing its efficacy in audio-visual question answering task.

## 5 CONCLUSION

In this work, we propose an effective Temporal-Spatial Perception Model framework for addressing complex question-answering task in dynamic audio-visual scenarios. It includes a temporal perception module with a declarative sentence text prompt and a spatial perception module incorporating token merging. These modules are employed to locate temporal segments relevant to the question and enhance spatial audio-visual associations, thereby facilitating fine-gained audio-visual scenes understanding. Extensive experiments demonstrate that the proposed framework achieves precise temporal-spatial perception on multiple benchmark, effectively showcasing the reasoning process involved in answering questions. We believe that our work will serve as inspiration researchers in the field of audio-visual scene understanding.

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
