# OpenReview forum: "Boosting Audio Visual Question Answering via Key Semantic-Aware Cues"
_acmmm.org/ACMMM/2024/Conference — MM2024 Poster_

### Official Review · Reviewer_39Hp · 2024-05-18

**Rating:** 4
**Confidence:** 2

**Summary:**

This paper proposes a method for audio-visual question answering, utilizing text prompts to guide the model in capturing temporal segments based on audio. The motivation sounds reasonable. The results surpass those of LAVISH, presented at CVPR 2023, and the ablation experiments validate the effectiveness of the results. However, there are some minor errors in the formulas and figures, and some parts of the description are unclear. See limitations.

**Strengths:**

1. The motivation of the paper is sound, addressing the limitation of audio-visual question answering tasks where non-declarative questions are hard to align with visual semantics.
2. The design of modules for query rewriting and the merging of similar tokens at the spatiotemporal level is quite innovative (though I am not fully aware of the latest advances in AVQA, so there might be some omissions).
3. The method achieved good results on two public AVQA datasets.
4. The code is provided, which should ensure reproducibility.

**Limitations:**

Overall quality is acceptable, but there are some areas I don’t quite understand. I'm not sure if it is due to my oversight:

1. How exactly is TPC performed? Is it manually designed? What are the design principles? How is the alignment between the rewritten TPrompt and visual semantics ensured? I believe this alignment is crucial for selecting key temporal segments, but the authors seem to have not provided a detailed explanation.
2. What are the dimensions of the aggregated feature $F_v$ in formula (5)? Why is it $F_p$ below it? If $f_p^{\lambda}$ and $F_v$ sizes are not consistent, it seems an equal sign cannot be used here.
3. In Section 3.4, line 514 mentions the activation function $\sigma$ and Tanh. Where are these used?
4. In line 548 of section 4.2: "we utilize the CLIP-ViT-L/14 model pre-trained on ImageNet." The authors should carefully check this statement.
5. VGGish is 128-d. Transforming 128-d to 512-d looks a bit odd; doing the reverse seems more usual. Is this approach consistent with other papers?
6. The top-k values for MUSIC-AVQA and AVQA are different. Is this related to the average length of the dataset videos?

**Suitability:**

3

---

### Official Review · Reviewer_66XT · 2024-05-20

**Rating:** 3
**Confidence:** 3

**Summary:**

This paper proposes an end-to-end method integrating audio and visual features for the Task of AVQA. Specifically, the authors first utilize a Text Prompt Constructor to transform the prompt to a declarative one, then design a Temporal Perception Module to identify the significant frames, lastly adopt a Spatial Perception Module with multimodal interaction to derive the final representation. The proposed model achieves great performances on the AVQA dataset, across Audio, Visual and Audio-Visual subtasks.

**Strengths:**

1.	This paper proposes an end-to-end approach to integrate multimodal features, i.e., audio and visual.
2.	This paper conducts comprehensive experiments on AVQA datasets, and performs extensive ablation studies to verify the effectiveness of each component.

**Limitations:**

1.	This paper overall seems like an add-on study of a previous work PSTP-Net[1]. Except for the Text Prompt Constructor, the author needs to specify the novelty and the difference between this work and [1]
2.	The authors did not give the concrete structure of TPC. The authors should specify how it is implemented. Sec 3.2 only gives a vague statement of “Leveraging the TPC, we manually transform the question into a declarative sentence”
3.	Some of the equations may not be properly written. For example, in Equation (5), why does $F_v$ only depends on $f_p^{\lambda}$ ?
4.	According to Table 3, the avg result achieves 73.35 even when all components are removed. This result beats or is comparable to almost all of the baselines in Table 1, including PSTP-Net. The authors may need to analyze this phenomenon.

[1] Li, G., Hou, W., & Hu, D. (2023, October). Progressive Spatio-temporal Perception for Audio-Visual Question Answering. In Proceedings of the 31st ACM International Conference on Multimedia (pp. 7808-7816).

**Suitability:**

3

---

### Official Review · Reviewer_113j · 2024-05-22

**Rating:** 5
**Confidence:** 3

**Summary:**

This paper proposes a new AVQA method that transforms non-declarative questions into declarative prompts to better identify relevant segments. While the idea is promising, the article lacks sufficient explanations for many key points, which need to be supplemented.

**Strengths:**

By constructing declarative prompts corresponding to the questions, it aligns well with the data format used by the CLIP model, which intuitively aids in segment matching. The motivation behind this approach is reasonable.
The experimental section is comprehensive, providing ample comparisons and evaluations.

**Limitations:**

The key methods lack specific explanations, and some of the explanations for certain questions are somewhat vague.

Questions:
1.	In the context of Figure 1, the mention of "the absence of objects' semantics in visual tokens" is not clearly expressed in the figure, and I don't quite understand it.
2.	Regarding the crucial TPC technique, only one example is provided, and there is no description of the method or process for constructing declarative sentence prompts based on a given question.
3.	Algorithm 3 mentions Φ without further explanation or reference.
4.	Step 3 in the process of merging visual tokens was not explained. If we follow the approach shown in Figure 3, why do we ultimately obtain an unequal number of a set tokens, b set tokens, and merge tokens
5.	Is there any other method of using prompt in the AVQA field? If there are, there is a lack of comparison between other methods and this article.

**Suitability:**

3

---

### Official Review · Reviewer_f12x · 2024-05-23

**Rating:** 2
**Confidence:** 3

**Summary:**

This paper focuses on audio-visual question answering problems, proposes a Temporal-Spatial Perception Model (TSPM), which is composed of Temporal Perception module and Spatial Perception module respectively. By enhancing features in specific time steps and spatial positions, it achieves better performance on AVQA tasks.

**Strengths:**

1. The model explicitly can intuitively locate and enhance features of specific time steps and spatial positions that have gain for the final performance.
2. The model achieves the best performance and surpasses the second most by a wide margin on MUSIC-AVQA.
3. The structure of the paper is clear and the writing is smooth.

**Limitations:**

1. The author did not provide technical details about the Text Prompt Constructor (TPC).
2. The Token Merge process disrupts the spatial order of the image tokens. Considering that this method uses a Transformer-based encoder, it is necessary for the authors to elaborate further on whether this process occurs before or after Position Encoding and its impact on sequence processing.
3. Algorithm 1 appears to be similar to the conventional top-k algorithm. Why is it listed separately?
4. Compared to PSTP-Net, this paper upgrades the visual and textual feature extractors from CLIP-ViT-B/32 to CLIP-ViT-L/14, which results in stronger representation capabilities and increased computation cost. Comparing their performance and parameters on this basis is unfair. The authors should either provide results using the same encoder or justify the necessity of this upgrade.

**Suitability:**

3

---

### Meta-Review · Area_Chair_zkLi · 2024-06-30

**Recommendation:** Accept (Poster)
**Confidence:** 5

**Metareview:**

The reviewers are satisfied with the authors' rebuttal and have no doubt that the paper quality is acceptable for the conference. The authors should incorporate the reviewers' insightful feedback including writing issues into their revision.

---

### Meta-Review · Senior_Area_Chairs · 2024-07-10

**Recommendation:** Accept (Poster)
**Confidence:** 5

**Metareview:**

This paper received mixed ratings initially. After rebuttal, all the reviewers tend to accept the paper. SAC and AC agree with reviewers and recommend acceptance.